# COVID-19 Severity and Survival over Time in Patients with Hematologic Malignancies: A Population-Based Registry Study

**DOI:** 10.3390/cancers15051497

**Published:** 2023-02-27

**Authors:** Joaquín Martínez-López, Javier De la Cruz, Rodrigo Gil-Manso, Adrián Alegre, Javier Ortiz, Pilar Llamas, Yolanda Martínez, José-Ángel Hernández-Rivas, Isabel González-Gascón, Celina Benavente, Pablo Estival Monteliu, Víctor Jiménez-Yuste, Miguel Canales, Mariana Bastos, Mi Kwon, Susana Valenciano, Marta Callejas-Charavia, Javier López-Jiménez, Pilar Herrera, Rafael Duarte, Lucía Núñez Martín-Buitrago, Pedro Sanchez Godoy, Cristina Jacome Yerovi, Pilar Martínez-Barranco, María García Roa, Cristian Escolano Escobar, Arturo Matilla, Belén Rosado Sierra, María Concepción Aláez-Usón, Keina Quiroz-Cervantes, Carmen Martínez-Chamorro, Jaime Pérez-Oteyza, Rafael Martos-Martinez, Regina Herráez, Clara González-Santillana, Juan Francisco Del Campo, Arancha Alonso, Adolfo de la Fuente, Adriana Pascual, Rosalía Bustelos-Rodriguez, Ana Sebrango, Elena Ruiz, Eriel Alexis Marcheco-Pupo, Carlos Grande, Ángel Cedillo, Carlos Lumbreras, Andrés Arroyo Barea, José Manuel Casas-Rojo, Maria Calbacho, José Luis Diez-Martín, Julio García-Suárez

**Affiliations:** 1Hematology Department, Hospital Universitario 12 de Octubre, imas12, Universidad Complutense, CNIO-ISCIII, CIBERONC, 28041 Madrid, Spain; 2Research Institute, Hospital Universitario 12 de Octubre, imas12, 28041 Madrid, Spain; 3Hematology Department, Hospital Universitario de La Princesa, IIS-HUP, 28006 Madrid, Spain; 4Hematology Department, Hospital Fundación Jiménez Díaz, Health Research Institute IIS-FJD, 28040 Madrid, Spain; 5Hematology Department, Hospital Universitario Infanta Leonor, 28031 Madrid, Spain; 6Hematology Department, Hospital Clínico San Carlos, 28040 Madrid, Spain; 7Hematology Department, Hospital Universitario La Paz, 28046 Madrid, Spain; 8Hematology Department, Clínica Universidad de Navarra, 28027 Madrid, Spain; 9Hematology Department, Hospital General Universitario Gregorio Marañón, Instituto de Investigación Sanitaria Gregorio Marañón, 28007 Madrid, Spain; 10Department of Medicine, Universidad Complutense, 28223 Madrid, Spain; 11Hematology Department, Hospital Universitario HM Sanchinarro, 28050 Madrid, Spain; 12Hematology Department, Hospital Universitario Príncipe de Asturias, Universidad de Alcalá, Alcalá de Henares, 28805 Madrid, Spain; 13Hematology Department, Hospital Universitario Ramón y Cajal, 28034 Madrid, Spain; 14Hematology Department, Hospital Universitario Puerta de Hierro Majadahonda, 28222 Madrid, Spain; 15Hematology Department, Hospital Universitario Severo Ochoa, 28911 Madrid, Spain; 16Hematology Department, Hospital Universitario Fundación Alcorcón, 28922 Madrid, Spain; 17Hematology Department, Hospital Universitario de Getafe, 28905 Madrid, Spain; 18Hematology Department, Hospital Central de la Defensa Gómez Ulla, 28047 Madrid, Spain; 19Hematology Department, Hospital Universitario Rey Juan Carlos, Móstoles, 28933 Madrid, Spain; 20Hematology Department, Hospital HLA Universitario Moncloa, 28027 Madrid, Spain; 21Hematology Department, Hospital Universitario de Móstoles, 28935 Madrid, Spain; 22Hematology Department, Hospital Universitario Quirónsalud Madrid, Pozuelo de Alarcón, 28223 Madrid, Spain; 23Hematology Department, Hospital Universitario General de Villalba, Villalba, 28400 Madrid, Spain; 24Hematology Department, Hospital Universitario Infanta Sofía, San Sebastián de Los Reyes, 28702 Madrid, Spain; 25Hematology Department, Hospital Universitario de Fuenlabrada, 28942 Madrid, Spain; 26Hematology Department, Hospital Universitario del Henares, Coslada, 28822 Madrid, Spain; 27Hematology Department, Hospital Ruber, 28006 Madrid, Spain; 28Hematology Department, MD Anderson Cancer Center Madrid, 28033 Madrid, Spain; 29Hematology Department, Hospital Universitario Infanta Elena, Valdemoro, 28340 Madrid, Spain; 30Hematology Department, Hospital Universitario del Sureste, Arganda del Rey, 28500 Madrid, Spain; 31Hematology Department, Hospital Universitario de Torrejón, 28850 Madrid, Spain; 32Hematology Department, Hospital Universitario del Tajo, Aranjuez, 28300 Madrid, Spain; 33Hematology Department, Hospital Universitario Infanta Cristina, Parla, 28980 Madrid, Spain; 34Asociación Madrileña de Hematología y Hemoterapia (AMHH), 28040 Madrid, Spain; 35Internal Medicine Department, Hospital Universitario 12 de Octubre, 28041 Madrid, Spain; 36Internal Medicine Department, Hospital Universitario Infanta Cristina, Parla, 28980 Madrid, Spain

**Keywords:** COVID-19, SARS-CoV-2, hematological malignancies, multiple myeloma, lymphoma, acute leukemia

## Abstract

**Simple Summary:**

There are contradictory data about coronavirus disease (COVID-19) in patients with hematological malignancies. In this population-based study we evaluated severity and survival of unvaccinated patients with hematological malignancies (HM) and COVID-19 in the Madrid region, Spain, between early February 2020 and February 2021. Also, a comparison was made with non-cancer patients from the SEMI-COVID registry and post COVID-19 conditions were evaluated. Overall, 30-day mortality was 32.7%, with higher mortality among certain groups of patients (aged ≥ 60 years, presence of ≥ 3 comorbidities, diagnosis of AML/ALL, treatment with conventional chemotherapy within 30 days of COVID-19 diagnosis, recipients of systemic corticosteroids as COVID-19 therapy). Mortality rates were similar between earlier and later phases of the pandemic, not paralleling the reduction of mortality in non-cancer patients. Up to 27.3% patients had a post COVID-19 condition. These findings will be useful to understand COVID-19 morbidity and mortality in unvaccinated patients diagnosed with HM.

**Abstract:**

Mortality rates for COVID-19 have declined over time in the general population, but data in patients with hematologic malignancies are contradictory. We identified independent prognostic factors for COVID-19 severity and survival in unvaccinated patients with hematologic malignancies, compared mortality rates over time and versus non-cancer inpatients, and investigated post COVID-19 condition. Data were analyzed from 1166 consecutive, eligible patients with hematologic malignancies from the population-based HEMATO-MADRID registry, Spain, with COVID-19 prior to vaccination roll-out, stratified into early (February–June 2020; *n* = 769 (66%)) and later (July 2020–February 2021; *n* = 397 (34%)) cohorts. Propensity-score matched non-cancer patients were identified from the SEMI-COVID registry. A lower proportion of patients were hospitalized in the later waves (54.2%) compared to the earlier (88.6%), OR 0.15, 95%CI 0.11–0.20. The proportion of hospitalized patients admitted to the ICU was higher in the later cohort (103/215, 47.9%) compared with the early cohort (170/681, 25.0%, 2.77; 2.01–3.82). The reduced 30-day mortality between early and later cohorts of non-cancer inpatients (29.6% vs. 12.6%, OR 0.34; 0.22–0.53) was not paralleled in inpatients with hematologic malignancies (32.3% vs. 34.8%, OR 1.12; 0.81–1.5). Among evaluable patients, 27.3% had post COVID-19 condition. These findings will help inform evidence-based preventive and therapeutic strategies for patients with hematologic malignancies and COVID-19 diagnosis.

## 1. Introduction

Despite increasing vaccine availability, the COVID-19 pandemic continues to cause substantial morbidity and mortality worldwide. As of 7 September 2022, 603,711,760 cases of COVID-19 had been reported, including 6,484,136 deaths [1]. Some western European countries saw three waves of COVID-19 before vaccination campaigns started in 2021. In Spain, the first wave commenced in February 2020 and lasted for 3 months before abating due to strict lockdown measures. Following relaxation of these measures, a second wave commenced in August 2020 and lasted until December 2020, and a third wave occurred in January–February 2021 [2,3]. These observations align with those for other western European countries [4,5,6]. However, trends over time in COVID-19 natural history, clinical course, and outcomes are not completely understood [7]. Empirical data suggest that mortality rates have declined over time, likely reflecting changes in transmission and case demographics, as well as advances in detection, prevention, and treatment [4,5,6,8]. Additionally, new coronavirus variants have resulted in differences in clinical characteristics and outcomes between waves [9].

Patients with hematologic malignancies (HM) have an elevated risk of developing severe and life-threatening infections because of immune deficiency and use of immunosuppressive treatments. Four large studies and a meta-analysis of 3377 HM patients with COVID-19 during the first wave reported a 33–40% mortality rate, with patients with acute myeloid leukemia (AML) at particularly high risk [10,11,12,13,14]. These rates were up to 10–40 times higher than those in the general population [15]. HM patients also had an increased mortality rate compared with patients with solid tumors (~1.7 times higher), independent of potential confounders such as performance status [16]. It is unclear if increased clinical experience and knowledge of COVID-19 management has changed outcomes in HM patients. Three studies compared trends in mortality between the first and the second/third waves of the pandemic. Data from an ongoing European Hematology Association (EHA) EPICOVIDEHA registry demonstrated a higher mortality rate in HM patients with COVID-19 during the first (40.7%) versus the second (24.8%) wave [17]. This is consistent with data on a cohort of chronic lymphoid leukemia (CLL) patients with COVID-19, which showed that mortality rate was 35% versus 11% in an earlier versus a later cohort [18]. In contrast, updated results on 656 patients from the American Society of Hematology (ASH) Research Collaborative Data Hub COVID-19 registry showed a continued high risk of death among HM patients with COVID-19, with a 33% mortality rate among patients hospitalized with COVID-19 [19,20]. Further studies are needed to clarify this important issue.

Approximately 50% of COVID-19 survivors have prolonged sequelae through 6 months post-recovery [21]. This is now termed post COVID-19 condition (PCC) by the World Health Organization (WHO) [22]. It has been speculated that PCC could be more frequent in cancer patients due to a variety of mechanisms including their impaired immune response [23]. However, the OnCovid European retrospective registry study of cancer patients with COVID-19 showed that, with a median post-COVID-19 follow-up period of 4 months, only 15% of 1557 COVID-19 survivor patients had sequelae (3–5 months from COVID-19 onset). Long-term sequelae (≥6 months from onset) were not assessed [23]. Importantly, this study demonstrated an association between PCC and worse survival. However, no studies looking at risk factors for PCC in a cancer patient population were identified.

HEMATO-MADRID COVID-19 is an ongoing prospective, observational, multicenter, population-based cohort study sponsored by the Madrid Society of Hematology (Asociación Madrileña de Hematología y Hemoterapia, AMHH, Spain), that was initiated in March 2020 with the aim of capturing longitudinal data regarding the consequences of COVID-19 for patient care and outcomes during and after the acute phase of COVID-19 in HM patients. Initial results from the first 697 patients were published in October 2020 [11]. The overall mortality rate was 33%, with the greatest risk observed among older patients, those with >2 comorbidities, AML patients, and those receiving active antineoplastic treatment with monoclonal antibodies. In this analysis, using an updated dataset with extended registration and follow-up periods, we aimed to identify independent prognostic factors for COVID-19 severity and survival. We compared patient mortality between those diagnosed with COVID-19 early (during the first wave) or later (during the second/third waves) and between unvaccinated HM inpatients and matched non-cancer COVID-19 inpatients. We also sought to describe the occurrence of and prognostic factors for PCC.

## 2. Material and Methods

### 2.1. Study Design and Participants

HEMATO-MADRID COVID-19 includes deidentified data on patients from 31 reporting healthcare centers with AMHH-affiliated hematologists throughout the Madrid region, Spain, covering 6.75 million inhabitants. The study methodology has been published previously [11]. The inclusion criteria were: age ≥ 18 years, diagnosis of SARS-CoV-2 infection confirmed by reverse transcription–polymerase chain reaction of a nasopharyngeal swab [24] taken from patients in emergency departments, hospital wards (patients infected while hospitalized), or outpatient clinics of participating centers, and a history of HM at any time, which could be active or in remission at the time of COVID-19 diagnosis. COVID-19 diagnosis was determined according to WHO international recommendations [25]. Decisions about hospitalization and intensive care unit (ICU) admission were made locally based on criteria that were updated daily during the pandemic. Patients were evaluated by investigators at each participating institution per local practice and when clinically indicated.

The study was funded by the Fundación Madrileña de Hematología y Hemoterapia and the Fundación Leucemia y Linfoma. The study protocol was approved by the Institutional Review Board (IRB) and Ethics Committee of the University Hospital 12 de Octubre, Madrid, Spain (Study ID: n#20/189) and then by the IRBs of all participating centers. Written informed consent was waived. The study was performed in accordance with the principles of the Declaration of Helsinki and the International Conference on Harmonization Good Clinical Practice guidelines.

### 2.2. Data Collection and End Points

Between 28 February 2020 and 18 February 2021, consecutive HM patients fulfilling the inclusion criteria were entered online into the registry by local investigators and records were updated through to 1 March 2021. There were no predefined time points for follow-up, and patients could be entered into the database at any time following their COVID-19 diagnosis. Database lock (1 March 2021) occurred prior to the roll-out of COVID-19 vaccinations to HM patients in Spain. The study steering committee, which includes members with expertise in HM and infectious diseases, reviewed each registered case for completeness and consistency. The key data extracted for the purposes of this analysis were pre-infection patient characteristics, cancer type and treatment, and information on COVID-19 management. Age, sex, and comorbidities associated with COVID-19 (cardiac disease, pulmonary disease not including lung cancer, renal disease, diabetes, hypertension, and body mass index (BMI) ≥ 35) were documented, along with type of HM and therapy received. A comorbidity count was determined based on presence of the six comorbidities prespecified above. ‘Active’ antineoplastic treatment was defined as patients having received anticancer therapy within 30 days prior to COVID-19 diagnosis; therapies were classified as conventional chemotherapy (includes intensive and standard dosing), low-intensity chemotherapy (includes Single-agent hydroxyurea, chlorambucil, or cyclophosphamide), hypomethylating agents (includes azacitidine and decitabine), monoclonal antibodies, immunomodulatory drugs (includes lenalidomide, pomalidomide, and thalidomide), molecular-targeted therapies (includes tyrosine kinase inhibitors, Bruton’s tyrosine kinase inhibitors, Aurora kinase inhibitors, PI3K inhibitors, proteasome inhibitors, and histone deacetylase inhibitors), or supportive care (includes transfusion and hematopoietic growth factor support).

The key end points of the study were clinical severity of COVID-19, mortality (30-day mortality and 30- and 60-day survival probability estimates), and PCC occurrence and characteristics. COVID-19 severity was assessed within 24 h of admission per WHO guidelines [26]. For analysis of prognostic factors for COVID-19 severity, patients were evaluated in mild/moderate and severe/critical groups. Patients were analyzed according to time of COVID-19 diagnosis; those diagnosed from 28 February through 30 June 2020, were defined as the early cohort (first wave); patients diagnosed from 1 July 2020 to 18 February 2021, were defined as the later cohort (second/third waves). Patient characteristics and cancer and COVID-19 management features were compared between cohorts. For comparisons with non-cancer COVID-19 inpatients, we used a reference cohort extracted from the SEMI-COVID registry run by the Spanish Society of Internal Medicine [27] that was propensity-score-matched to HM inpatients by age and sex. The comparison was restricted to Madrid region resident inpatients. PCC symptoms were evaluated at 4- and 6-months post diagnosis; patients with any PCC-associated symptoms at either time point were classified as having PCC.

### 2.3. Statistical Analyses

Baseline characteristics and cancer features were described by type of hematological malignancy and clinical severity level (mild-moderate and severe-critical). Strength of association between each potential prognostic factor and COVID-19 severity was estimated with logistic regression models; multivariate analyses with age, sex, and number of comorbidities as covariates provided adjusted odds ratios (ORs) and 95% confidence intervals (CI). For each prognostic factor, 30- and 60-day survival probabilities were estimated using the life-table method, with hazard ratios (HRs) and 95% CI determined using Cox proportional-hazard regression models with age, sex, and comorbidity count as covariates.

Patient characteristics and cancer and COVID-19 management features were compared between study periods. Non-cancer inpatients were propensity-score-matched on age, sex, and region with HM inpatients. Due to interoperability limitations of the AMHH and SEMI-COVID registries, mortality was compared in terms of proportions at 30-day.

Prognostic factors for PCC were assessed in multivariate logistic regression models. Missing data were reported for each prognostic variable by outcome. Analyses were generated using SAS/STAT software, Version 9.4, SAS Institute Inc., Cary, NC, USA.

## 3. Results

### 3.1. Patients

Thirty-one hospitals, covering 98% of Madrid region population, Spain, reported cases of HM patients with COVID-19 for potential inclusion in this analysis (Figure 1). Of 1408 cases reported, 1166 met the inclusion criteria. Among these 1166 patients, the median age was 71 years (interquartile range, 59–79 years), 59.7% were male, and 41.5% and 31.6% had 1 or ≥2 of the six specified COVID-19-associated comorbidities (Table 1). Overall, 839 (72.0%) patients had a lymphoid malignancy and 327 (28.0%) had a myeloid malignancy. The most common HMs were non-Hodgkin’s lymphoma (NHL, *n* = 325 [27.9%]), multiple myeloma (MM, *n* = 263, 22.6%), CLL (*n* = 175, 15.0%), myelodysplastic syndrome (MDS, *n* = 115, 9.9%), and AML (*n* = 92, 7.9%) (Table 1).

In total, 679/1162 (58.4%) patients with known treatment had received antineoplastic treatment within 30 days before COVID-19 diagnosis, the most common being conventional chemotherapy (*n* = 260, 22.4%), molecular-targeted therapy (*n* = 130, 11.2%), immunomodulator drugs, and low-intensity chemotherapy (each *n* = 71, 6.1%) (Table 1). A known history of stem cell transplantation (SCT) was reported for 156/1127 (13.8%) patients, with 100 (8.9%) and 56 (5.0%) having received autologous and allogeneic SCT, respectively; the median age of these patients was 60.0 years (IQR 49.0–66.0), and the median time since transplantation was 22.0 months (IQR 9.0–56.0).

### 3.2. COVID-19 Diagnosis and Treatment

Of 1166 HM patients, 896 (76.8%) were hospitalized and 270 (23.2%) received only ambulatory management. Among inpatients, regardless of outcome, median hospital stay was 14 days (IQR, 6–40), and 273 (30.6%) were treated in an ICU. In total, 769 (66.0%) HM patients were included in the early cohort and 397 (34.0%) were included in the later cohort. Appendix A (Appendix A) summarizes patient characteristics and COVID-19 management by cohort. A lower proportion of patients were hospitalized in the later waves (54.2%) compared to the earlier (88.6%), OR 0.15, 95%CI 0.11–0.20. The overall proportion of patients admitted to an ICU was similar between cohorts (22.1% vs. 25.9%, OR 1.25; 95% CI: 0.94–1.66). The proportion of hospitalized patients admitted to the ICU was higher in the later cohort (103/215, 47.9%) compared with the early cohort (170/681, 25.0%, OR 2.77; 95% CI: 2.01–3.82). COVID-19 treatments also differed between the early and later cohorts, with significantly lower rates of antivirals (86.0% vs. 21.4%, OR 0.04; 95% CI: 0.03–0.06) and tocilizumab (17.4% vs. 10.3%, OR 0.55; 95% CI: 0.38–0.79) use in the later cohort. Corticosteroids were widely used in both cohorts, but at a higher rate in the later (58.9%) vs. earlier (51.0%) cohort (OR 1.38; 95% CI: 1.08–1.76) (Appendix A).

### 3.3. Comparison between the Early and Later Cohorts

HM patients in the later cohort were younger (mean difference, 2.4 years; 95% CI: 0.6–4.2) and had fewer comorbidities than those in the early cohort: 32.0% vs. 24.3% (OR, 0.68; 95% CI: 0.52–0.89) had none of the six COVID-19-related comorbidities, and 65.2% vs. 48.0% (OR 0.49; 95% CI: 0.38–0.63) had no other comorbidity (Appendix A). A larger proportion of HM patients in the later cohort versus the early cohort had received conventional chemotherapy within 30 days of diagnosis (27.5% vs. 19.6%, OR 1.55; 95% CI: 1.17–2.06).

Mortality risk did not differ significantly between cohorts: 30-day survival was 67.4% vs. 70.9% in the early versus later cohort, and 60-day survival was 56.3% vs. 55.8% (adjusted HR 0.99, 95% CI: 0.79–1.26; Table 2).

In an analysis restricted to inpatients, the proportions of patients with critical COVID-19 (151/677, 22.3%, vs. 64/209, 30.6%, OR 1.54; 95% CI: 1.09–2.17), receiving high-flow oxygen support or mechanical ventilation (163/678, 24.0%, vs. 100/214, 46.7%, OR 2.77; 95% CI: 2.01–3.82), and admitted to an ICU were lower in the early versus later cohort. After adjustment for HM subtype and therapy, inpatient mortality risk did not differ between the later versus early cohorts (HR 1.26, 95% CI: 0.98–1.61).

### 3.4. Factors Associated with COVID-19 Severity

Data on COVID-19 clinical severity were available for 1131 patients; 508 (44.9%) had mild/moderate and 623 (55.1%) severe/critical disease (Table 3). The proportion of patients aged ≥ 60 years was higher in the severe/critical (521/621, 83.9%) versus mild/moderate (313/491, 63.7%, OR 2.96; 95% CI: 1.23–3.93) group. Similarly, the proportion of patients with ≥2 COVID-19-associated comorbidities was higher in the severe/critical (36.8%) versus mild/moderate (25.2%; OR 1.62; 95% CI: 1.14–2.30) group (Table 3). The proportion of patients with severe/critical COVID-19 was higher in the early (470/760, 61.8%) versus later cohort (153/371, 41.2%; OR 2.31; 95% CI: 1.79–2.97) cohort; among inpatients, there was no difference in the proportion with severe/critical COVID-19 between the early and later cohorts (470/677, 69.4%, and 153/209, 73.2%, respectively, OR 0.83; 95% CI: 0.59–1.18).

After adjusting for age, sex, and comorbidities, age ≥ 60 years (OR 2.52, 95% CI: 1.87–3.39), ≥1 comorbidity (1.68, 1.26–2.24), and having AML (3.13, 1.83–5.34), acute lymphoblastic leukemia (ALL; 2.88, 1.22–6.82), or CLL (2.02, 1.34–3.05) were independently associated with severe/critical COVID-19. Receipt of any specific active antineoplastic therapy or history of SCT were not associated with COVID-19 severity (Table 3).

### 3.5. Factors Associated with Mortality

At data cut-off, with a median follow-up of 40 days (IQR, 16–99), 381 of 1166 (32.7%) patients had died. The 30-day and 60-day survival probabilities were 68.4% (95% CI: 65.3–71.3) and 56.3% (52.6–59.9), respectively (Table 2). Kaplan–Meier survival estimates by COVID-19 severity and time of diagnosis are shown in Figure 2.

Table 2 summarizes 30-day and 60-day survival probabilities, and HRs, by patient-, HM-, and COVID-19-related characteristics and features. Kaplan–Meier survival estimates by HM type, treatment status, type of cancer therapy, and SCT history are shown in Appendix A. On multivariate analysis, after adjusting for age, sex, and comorbidities, mortality risk was higher in patients aged ≥ 60 years (HR 2.40, 95% CI: 1.71–3.38) and with ≥3 comorbidities (1.43, 1.01–2.03). Patients with ALL (HR 2.31, 95% CI: 1.04–5.12) or AML (1.68, 1.17–2.40) had an increased mortality risk compared with patients with NHL; similarly, patients treated with conventional chemotherapy within 30 days of COVID-19 diagnosis had an increased mortality risk compared with patients receiving no active treatment (HR 1.49, 95% CI: 1.14–1.93). Patients with a history of autologous SCT had a lower mortality risk than those with no history of transplant (HR 0.54, 95% CI: 0.31–0.95). HM patients who received systemic corticosteroid therapy for COVID-19 had an increased mortality risk (HR 2.06, 95% CI: 1.64–2.59).

### 3.6. Comparison between HM and Non-Cancer Inpatients with COVID-19

In a contemporaneous reference cohort of non-cancer inpatients extracted from the Spanish SEMI-COVID registry, overall mortality during the study period was 19.4% (*n* = 3845/19,813). Compared to this cohort, HM inpatients in HEMATO-MADRID COVID-19 were older (mean difference, 3.9 years, 95% CI: 2.8–5.0) and more commonly male (OR 1.14, 95% CI: 0.99–1.31). Similar differences were seen when the comparison was limited to SEMI-COVID registry non-cancer inpatients in the Madrid region (Table 4); HM inpatients were thus matched by age and sex with non-cancer inpatients for further comparisons.

Overall, 669 and 207 HM inpatients and propensity-score-matched non-cancer inpatients were analyzed from the early and later cohorts, respectively (Table 4). Across cohorts, HM inpatients had fewer comorbidities than non-cancer inpatients, except for renal disease (109/876, 12.4%, vs. 48/876, 5.5%; OR 2.45, 95% CI: 1.72–3.49), they were more likely to have been treated with tocilizumab (172/876, 19.6%, vs. 79/876, 9.0%, OR 2.46; 95% CI: 1.85–3.28) and corticosteroids (474/876, 54.1%, vs. 354/876, 40.4%, OR 1.74; 95% CI: 1.44–2.10) than non-cancer inpatients, and they were more likely to have received high-flow oxygen support or mechanical ventilation (258/876, 29.5%, vs. 175/876, 20.0%, OR 1.67; 95% CI: 1.34–2.08). The 30-day mortality differential between HM and non-cancer inpatients was significantly changed between the early and later cohorts (Breslow–Day test for homogeneity of odds ratio, *p* < 0.0001). In the early cohort, 30-day mortality was 32.3% and 29.6% for HM and non-cancer inpatients, respectively (OR 1.13, 95% CI: 0.90–1.43), but in the later cohort, 30-day mortality was higher in HM versus non-cancer patients (34.8% vs. 12.6%; OR 3.71, 95% CI: 2.25–6.13). The decrease in 30-day mortality in the later vs. early cohort in non-cancer patients (OR 0.34, 95% CI: 0.22–0.53) was not paralleled in HM inpatients (1.12, 0.81–1.55).

### 3.7. Post COVID-19 Condition

PCC data were available for 278/1166 (23.8%) patients; patients who died before (*n* = 324, 27.8%), or were not followed until (*n* = 404, 34.6%), 12 weeks from COVID-19 diagnosis, and patients whose COVID-19 diagnosis was less than 12 weeks from data cut-off (*n* = 160, 13.7%), were excluded. Of 278 patients assessed for symptoms at 4 and/or 6 months, 63 (22.7%) reported symptoms at 4 months and 49 (17.6%) at 6 months, resulting in 76 (27.3%) being classified as having PCC. Among these patients, respiratory symptoms were reported by 44 (57.9%) and asthenia by 29 (38.2%). Associations between patient-related, cancer-related, and COVID-19-related factors and PCC are summarized in Table 5; independent prognostic factors for PCC included having: a comorbidity of respiratory disease; been hospitalized; had critical COVID-19; or been treated with hydroxychloroquine, lopinavir/darunavir, remdesivir, or tocilizumab. Patients treated with corticosteroids or high-flow oxygen support or mechanical ventilation were also more likely to develop PCC.

## 4. Discussion

This analysis of a large cohort of unvaccinated HM patients with COVID-19, with comprehensive prospective data collection, provides valuable information on mid-term outcomes such as 60-day mortality, on clinical characteristics and mortality rates in those diagnosed with COVID-19 in the first versus later waves, and on differences between HM inpatients and propensity-score-matched non-cancer inpatients. Importantly, our patient series is highly representative of the overall population as we collected data on an unselected patient population, including outpatients.

Reporting both short-term end points and mid-term outcomes is extremely important in the setting of HM care, but, to our knowledge, there have been no previous studies that have reported data on 60-day mortality of HM patients with COVID-19. Through our prolonged follow-up, we demonstrated a 12.1-percentage-point increase in mortality between day 30 and day 60 following COVID-19 diagnosis. The increase in mortality observed after 30 days may reflect the deep immunosuppression associated with most HM subtypes and their treatment.

We also identified independent predictors of increased mortality risk, including age ≥ 60 years, presence of ≥ 3 comorbidities, and acute leukemia (AML and ALL). These findings reflect risk factors observed in other studies of HM patients [11,12,13,14,15,16]. Additionally, the impact of certain cancer treatments on COVID-19-related mortality is a question of paramount importance for hematologists. In our cohort, receipt of systemic conventional chemotherapy within 30 day of COVID-19 diagnosis was associated with an increased mortality risk versus not having received active therapy, but mortality risk was not significantly increased by receipt of monoclonal antibody therapy or molecular-targeted therapy.

Other data regarding the impact of chemotherapy on outcomes in HM patients with COVID-19 are inconsistent, with some studies identifying an association between recent chemotherapy and adverse outcomes [16,28,29,30,31]. and others not [11,12,13]. A meta-analysis [14] showed no association between mortality rate in HM patients with COVID-19 and recent chemotherapy. However, the analysis should be interpreted with caution given the heterogeneous definitions used for ‘recent treatment’ among the studies included. Moreover, differences in underlying disease and disease severity, as well as therapy intensity, were not fully accounted for. Clinicians should make treatment decisions on a case-by-case basis with their patients, taking into consideration the regional prevalence of COVID-19 and availability of healthcare resources.

Regarding COVID-19 treatment, we also found that HM patients who received systemic corticosteroid therapy for COVID-19 had an increased mortality risk, a finding consistent with a study of CLL patients [18]. Data from the RECOVERY trial supporting use of dexamethasone in hospitalized patients were published in July 2020 [32], and consequently 58.9% of HM patients in our later cohort received corticosteroids, including 92.3% of inpatients. The inferior survival in these patients may be due to corticosteroids being used mostly in patients with more severe disease. While these data are not sufficient to warrant a change in recommendations for corticosteroid use, given the demonstrated benefit in RECOVERY, they raise questions about corticosteroid benefit in HM patients with COVID-19.

Moreover, we found no difference in survival between HM patients diagnosed with COVID-19 in the first or second/third waves of the pandemic. This is in contrast to some studies in similar populations that reported a decline in mortality over time [17,18], suggested explanations for which include better healthcare organization, changes in patients’ characteristics, detection of more asymptomatic/mild cases, and improved treatments against COVID-19. In particular, the inclusion of less severe cases could contribute to lower mortality, a point not always analyzed in the literature. A shorter follow-up time in other studies could also contribute to a lower mortality. In our study, we found some differences in patient characteristics between the later and early cohorts that may help explain the lack of survival difference between waves. Although the proportion of ICU admissions was similar in both total cohorts, among hospitalized HM patients, the proportions of patients with critical COVID-19 and of ICU admissions was higher in the later cohort. More HM patients received conventional chemotherapy during the second/third waves. It has been reported that HM patients receiving chemotherapy have higher SARS-CoV-2 viral loads and that this correlates with in-hospital mortality [33].

Our comparison with contemporaneous, matched non-cancer inpatients with COVID-19 demonstrated that HM inpatients with COVID-19 were less likely to have comorbidities such as cardiac disease and hypertension, but more frequently received high-flow oxygen support or mechanical ventilation. While inpatient 30-day mortality during the first wave was similar between HM and non-cancer patients, the mortality rate during the second/third waves was almost three times higher among HM inpatients. This finding does not appear to be explained by differences in pharmacologic therapies received for COVID-19.

To our knowledge, this is one of the largest cohort studies, with the longest duration of follow-up, assessing PCC in HM patients. We found that 27.3% of HM patients experienced post-COVID-19 symptoms, most commonly respiratory symptoms and asthenia. Notably, 17.6% of patients reported post-COVID-19 symptoms 6 months post diagnosis. These rates are higher than reported for cancer patients in the OnCovid study after a 4-month follow-up period (15%) [23] and similar to findings in patients with chronic myeloproliferative disorders in a European LeukemiaNet observational study (30%) [34]. HM patients with PCC were ≥2-fold more likely to have a respiratory comorbidity, to have been hospitalized, or to have received antivirals or anti-inflammatory agents for severe COVID-19 than those without PCC. Anti-inflammatory agents such as corticosteroids may delay the elimination of the virus and contribute to protracted symptoms, especially in those with impaired immune systems [35,36].

None of our HM patients had received COVID-19 vaccines at the time of this study. Widespread vaccine administration is a cornerstone for controlling the pandemic and lowering mortality rates. In Spain, the vaccination program in cancer patients began on 15 March 2021. In the context of the current vaccination roll-out, further analysis of HEMATO-MADRID COVID-19 to explore the consequences of vaccination on mortality rates is of critical importance and will be undertaken in the near future.

Some limitations of our study relate to the voluntary reporting system of the registry by hematology hospital departments. Patients in the registry may have intersected more frequently with the medical system than the general population of patients with hematology malignancies. We provided outcome data by active cancer treatment and ambulatory/hospitalization management status to inform on a potential selection bias.

We intentionally limited multivariate adjustments to the covariates age, sex and comorbidity count due to the descriptive prognostic nature of the analyses. The post COVID-19 condition analysis lacked a comparison with the general population because we did not find suitable source of data. Even though our study includes only unvaccinated patients, with expected changes on morbidity and mortality trends after vaccination roll-out, it is still relevant to reiterate the elevated risk of severe COVID-19 among this population, since determined groups of patients (as recipients of SCT) may not be able to be vaccinated for a certain period of time.

## 5. Conclusions

In conclusion, despite increased clinical experience and knowledge regarding the management of this challenging infectious disease, as well as changes in therapeutic approaches, we did not find significant differences in mortality in HM patients between COVID-19 waves, neither in our cohort overall nor among inpatients, after accounting for several confounding factors. This study reiterates the increased risk of severe/critical COVID-19 and related death among HM patients, specifically those who are aged ≥ 60 years, have ≥3 comorbidities, have AML/ALL, have recently received cytotoxic therapy, or have received systemic corticosteroid therapy for COVID-19. PCC affects up to 27% of HM patients after recovery from COVID-19. This should be considered in public health policies targeted at protecting this vulnerable group. With better understanding of the clinical course of COVID-19 in this population, we can develop evidence-based preventive and therapeutic strategies for HM patients during the current pandemic and similar future healthcare challenges.

## Figures and Tables

**Figure 1 cancers-15-01497-f001:**
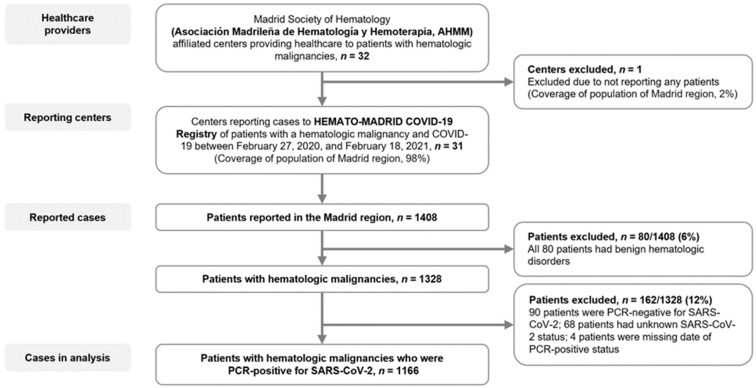
Flow diagram. Patients with hematologic malignancies who were reported as having COVID-19 and who were included in the present analysis. Reporting hospitals included 25/26 Madrid regional health service centers (8/8 designated high complexity level hospitals (CLH); 12/12 intermediate CLH; 5/6 low CLH), and 6/6 private centers.

**Figure 2 cancers-15-01497-f002:**
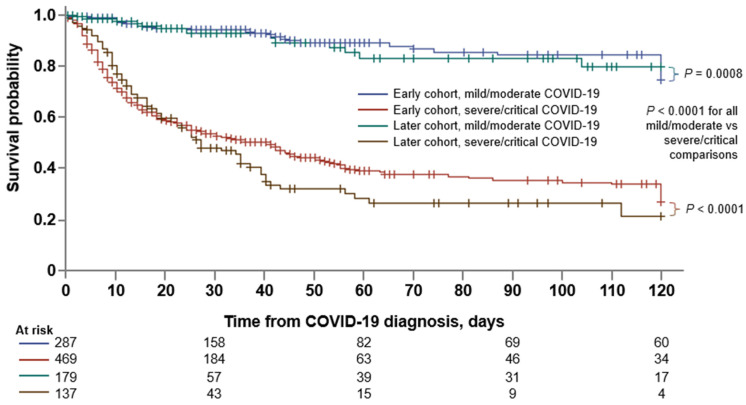
Survival estimates according to period of COVID-19 diagnosis and clinical severity of disease. Kaplan–Meier estimates of survival from time of COVID-19 diagnosis according to the period in which patients were diagnosed (early cohort vs. later cohort) and the clinical severity of COVID-19 (mild/moderate vs. severe/critical). *p* values by log rank test, with Šidák correction for multiple comparison.

**Table 1 cancers-15-01497-t001:** Baseline characteristics of, and therapy received by, patients with hematologic malignancies and COVID-19.

	Patients with Hematologic Malignancies, *N* = 1166									
	Total	Lymphoid Malignancies					Myeloid Malignancies			
		NHL	MM	CLL	HL	ALL	MDS	AML	CML	Ph-MPN
Patients, *n* (%)	1166 (100)	325 (27.9)	263 (22.6)	175 (15.0)	50 (4.3)	26 (2.2)	115 (9.9)	92 (7.9)	33 (2.8)	87 (7.5)
Age, y	*N* = 1147	*n* = 322	*n* = 253	*n* = 174	*n* = 49	*n* = 26	*n* = 113	*n* = 91	*n* = 32	*n* = 87
Median (IQR)	71 (59–79)	69 (57–76)	72 (63–79)	74 (63–82)	57 (42–71)	47 (33–59)	79 (71–84)	65 (50–75)	64 (54–84)	72 (65–82)
Sex, *n* (%)	*N* = 1152	*n* = 321	*n* = 261	*n* = 175	*n* = 49	*n* = 26	*n* = 113	*n* = 91	*n* = 33	*n* = 83
Female	464 (40.3)	136 (42.4)	108 (41.4)	61 (34.9)	18 (36.7)	8 (30.8)	41 (36.3)	46 (50.5)	8 (24.2)	28 (34)
Male	688 (59.7)	185 (57.6)	153 (58.6)	114 (65.1)	31 (63.3)	18 (69.2)	72 (63.7)	45 (49.5)	25 (75.8)	55 (66)
Comorbidity count	*N* = 1166	*n* = 325	*n* = 263	*n* = 175	*n* = 50	*n* = 26	*n* = 115	*n* = 92	*n* = 33	*n* = 87
0	314 (26.9)	96 (29.4)	62 (23.6)	44 (25.1)	13 (26.0)	15 (57.7)	22 (19.1)	33 (35.9)	11 (33.3)	18 (20.7)
1	484 (41.5)	140 (43.1)	102 (38.8)	77 (44.0)	18 (36.0)	8 (30.8)	50 (43.5)	40 (43.5)	9 (27.3)	40 (46.0)
≥2	368 (31.6)	89 (27.4)	99 (37.6)	54 (30.9)	19 (38)	3 (11.6)	43 (37.4)	19 (20.7)	13 (39.4)	29 (33.3)
Stem cell transplantation, * *n* (%)	*N* = 1127	*n* = 312	*n* = 259	*n* = 168	*n* = 47	*n* = 24	*n* = 112	*n* = 92	*n* = 32	*n* = 81
Autologous	100 (8.9)	23 (7.4)	66 (25.5)	0	7 (14.9)	1 (4.2)	1 (0.9)	2 (2.2)	0	0
Allogeneic	56 (5.0)	5 (1.6)	3 (1.2)	1 (0.6)	4 (8.5)	9 (37.5)	9 (8.0)	25 (27.2)	0	0
No	971 (86.2)	284 (91.0)	190 (73.4)	167 (99.4)	36 (76.6)	14 (58.3)	102 (91.1)	65 (70.7)	32 (100)	81 (100)
Cancer therapy, within 30 d, ^†^ *n* (%)	*N* = 1162	*n* = 325	*n* = 260	*n* = 175	*n* = 50	*n* = 26	*n* = 115	*n* = 92	*n* = 33	*n* = 86
Active therapy	679 (58.4)	178 (54.8)	190 (73.1)	55 (31.4)	24 (48.0)	16 (61.5)	50 (43.5)	61 (66.3)	29 (87.9)	76 (88.4)
Conventional chemotherapy	260 (22.4)	112 (34.5)	82 (31.5)	3 (1.7)	17 (34.0)	13 (50.0)	2 (1.7)	31 (33.7)	0	0
Low-intensity chemotherapy	71 (6.1)	4 (1.2)	8 (3.1)	3 (1.7)	0	1 (3.8)	3 (2.6)	0	4 (12.1)	48 (55.8)
Molecular-targeted therapy	130 (11.2)	13 (4.0)	20 (7.7)	43 (24.6)	0	1 (3.8)	1 (0.9)	5 (5.4)	24 (72.7)	23 (26.7)
Immunotherapy, mAb only ^‡^	56 (4.8)	38 (11.7)	10 (3.8)	3 (1.7)	4 (8.0)	1 (3.8)	0	0	0	0
Immunomodulator drugs	71 (6.1)	1 (0.3)	69 (26.5)	0	0	0	1 (0.9)	0	0	0
Hypomethylating agents	49 (4.2)	1 (0.3)	0	0	0	0	24 (20.9)	24 (26.1)	0	0
Supportive therapy	31 (2.7)	5 (1.5)	0	2 (1.1)	0	0	19 (16.5)	0	0	5 (5.8)
Active, not detailed	11 (0.9)	4 (1.2)	1 (0.4)	1 (0.6)	3 (6.0)	0	0	1 (1.1)	1 (3.0)	0
No active therapy	483 (41.6)	147 (45.2)	70 (26.9)	120 (68.6)	26 (52.0)	10 (38.5)	65 (56.5)	31 (33.7)	4 (12.1)	10 (11.6)

ALL, acute lymphoid leukemia; AML, acute myeloid leukemia; BMI, body mass index; CLL, chronic lymphocytic leukemia; CML, chronic myeloid leukemia; HL, Hodgkin’s lymphoma; IQR, interquartile range; MDS, myelodysplastic syndrome; mAb, monoclonal antibody; MM, multiple myeloma; NHL, non-Hodgkin’s lymphoma; Ph-MPN, Philadelphia chromosome-negative myeloproliferative neoplasm; * Patients who had ever received an autologous or allogeneic stem cell transplant. ^†^ Cancer therapy received within 30 days before COVID-19 diagnosis date. ^‡^ Includes single-agent anti-CD20 agents, daratumumab, and others.

**Table 2 cancers-15-01497-t002:** Kaplan–Meier estimates of 30-day and 60-day survival in patients with hematologic malignancies and COVID-19.

	Survival Estimate, % (95% CI)		Hazard Ratio (95% CI)	
	30-d	60-d	Unadjusted	Adjusted *
All patients ^†^	68.4 (65.3–71.3)	56.3 (52.6–59.9)		
Age				
Age 18–49 y	93.7 (86.6–97.1)	90.1 (80.8–95.1)	reference	reference
Age 50–79 y	72.8 (68.9–76.2)	58.8 (54.0–63.3)	3.59 (2.01–6.42)	3.01 (1.67–5.43)
Age ≥ 80 y	46.3 (39.8–52.5)	35.1 (28.3–42.0)	7.86 (4.36–14.2)	6.24 (3.39–11.49)
Sex				
Female	69.5 (64.5–74.0)	60.0 (54.2–65.4)	reference	reference
Male	67.4 (63.3–71.2)	53.6 (48.6–58.3)	1.12 (0.91–1.38)	1.15 (0.93–1.41)
Comorbidities				
0	79.3 (73.5–84.0)	69.7 (62.3–76.0)	reference	reference
1	70.9 (66.1–75.1)	56.3 (50.4–61.7)	1.49 (1.12–1.98)	1.10 (0.83–1.48)
2	61.4 (53.7–68.2)	49.7 (41.0–57.9)	1.89 (1.37–2.61)	1.15 (0.82–1.61)
≥3	50.4 (41.4–58.7)	41.6 (32.0–51.0)	2.57 (1.84–3.59)	1.43 (1.01–2.03)
Hematologic malignancy				
NHL	71.0 (64.9–76.2)	57.2 (49.9–63.8)	reference	reference
MM	69.7 (62.9–75.5)	57.3 (49.1–64.7)	0.98 (0.73–1.32)	0.83 (0.61–1.12)
CLL	59.1 (50.5–66.8)	49.8 (39.8–59.0)	1.37 (1.00–1.87)	1.02 (0.74–1.41)
HL	73.0 (55.6–84.4)	68.1 (49.0–81.3)	0.77 (0.43–1.41)	0.89 (0.49–1.64)
ALL	78.4 (52.2–91.3)	47.0 (17.6–72.1)	0.99 (0.46–2.13)	2.31 (1.04–5.12)
MDS	59.4 (48.9–68.4)	48.7 (37.2–59.3)	1.46 (1.04–2.05)	0.96 (0.68–1.37)
AML	63.8 (52.2–73.2)	53.6 (41.3–64.3)	1.41 (0.99–2.01)	1.68 (1.17–2.40)
CML	81.8 (58.5–92.8)	81.8 (58.5–92.8)	0.58 (0.24–1.43)	0.44 (0.18–1.08)
Ph-MPN	82.4 (71.0–89.6)	66.4 (50.9–78.0)	0.70 (0.43–1.13)	0.51 (0.31–0.82)
Stem cell transplantation ^‡^				
No	65.7 (62.2–68.9)	53.9 (49.8–57.8)	reference	reference
Autologous	85.3 (75.1–91.6)	80.8 (68.7–88.6)	0.58 (0.34–0.99)	0.54 (0.31–0.95)
Allogeneic	86.2 (71.8–93.6)	69.6 (51.2–82.2)	0.35 (0.20–0.59)	1.15 (0.64–2.07)
Cancer therapy, within 30 d ^¶^				
No active therapy	67.4 (62.3–71.9)	58.7 (51.8–63.3)	reference	reference
Active therapy	69.0 (65.0–72.7)	55.4 (50.6–60.1)	1.08 (0.88–1.33)	1.10 (0.89–1.36)
Conventional chemotherapy	66.4 (59.6–72.3)	51.8 (44.0–59.2)	1.16 (0.90–1.50)	1.49 (1.14–1.93)
Low-intensity chemotherapy	76.8 (63.4–85.8)	63.1 (46.1–76.1)	0.80 (0.48–1.32)	0.65 (0.39–1. 08)
Molecular targeted therapy	69.2 (59.4–77.1)	60.4 (48.9–70.0)	1.05 (0.75–1.47)	1.02 (0.73–1.44)
Immunotherapy, mAb only	68.9 (53.2–80.2)	47.7 (30.2–63.2)	1.14 (0.72–1.80)	1.40 (0.88–2.24)
Immunomodulator drugs	70.4 (56.8–80.5)	58.2 (42.3–71.1)	0.88 (0.56–1.39)	0.84 (0.53–1.33)
Hypomethylating agents	64.6 (47.6–77.3)	60.5 (42.8–74.3)	1.46 (0.94–2.27)	1.07 (0.68–1.67)
Supportive therapy	64.7 (43.1–79.8)	41.2 (19.9–61.4)	1.39 (0.79–2.46)	0.90 (0.51–1.61)
Time period of COVID-19 diagnosis				
Early cohort (1st wave, February–June 2020)	67.4 (63.7–70.8)	56.3 (52.0–60.4)	reference	reference
Later cohort (2nd/3rd wave, July 2020–February 2021)	70.9 (64.9–76.1)	55.8 (48.0–62.9)	0.93 (0.73–1.17)	0.99 (0.79–1.26)
Care setting of COVID-19 treatment				
Outpatient	99.4 (95.5–99.9)	93.8 (85.3–97.4)	reference	reference
Inpatient	62.1 (58.6–65.4)	49.3 (45.2–53.2)	10.8 (5.37–21.8)	8.81 (4.37–17.8)
Intensive care unit	45.2 (38.9–51.2)	28.0 (22.1–34.1)	2.21 (1.80–2.71)	2.42 (1.97–2.99)
Pharmacologic therapies for COVID-19 ^¶¶^				
(Hydroxy)chloroquine	67.7 (63.6–71.4)	56.9 (52.2–61.2)	1.00 (0.81–1.23)	0.96 (0.77–1.19)
Azithromycin	67.7 (62.5–72.5)	56.6 (50.3–62.3)	1.06 (0.86–1.30)	1.03 (0.83–1.27)
Lopinavir/darunavir	66.1 (60.9–70.7)	56.1 (50.3–61.5)	1.03 (0.84–1.26)	1.09 (0.88–1.35)
Remdesivir	80.5 (68.2–88.4)	61.5 (46.3–73.6)	0.71 (0.46–1.08)	0.84 (0.55–1.28)
Tocilizumab	66.3 (58.3–73.2)	52.0 (43.1–60.1)	1.07 (0.83–1.37)	1.25 (0.96–1.62)
Corticosteroids	59.8 (55.4–63.8)	45.2 (40.4–49.9)	2.15 (1.71–2.69)	2.06 (1.64–2.59)
Oxygen support during COVID-19 treatment				
No	98 (94–99)	91 (85–95)	reference	reference
Low-flow oxygen support	67.5 (63.0–71.6)	57.7 (52.4–62.6)	4.9 (3.09–7.62)	3.68 (2.33–5.81)
High-flow oxygen support or mechanical ventilation	44.3 (37.9–50.4)	26.8 (20.9–33.0)	10.1 (6.4–15.9)	8.52 (5.38–13.5)
Clinical severity of COVID-19				
Mild	96.1 (91.6–98.2)	90.5 (82.3–95.7)	reference	reference
Moderate	92.9 (88.2–95.7)	85.8 (78.9–90.6)	1.45 (0.77–2.74)	1.17 (0.62–2.22)
Severe	57.8 (52.5–62.8)	46.8 (40.8–52.5)	7.34 (4.26–12.7)	5.64 (3.26–9.79)
Critical	40.9 (34.0–47.7)	21.7 (15.7–28.4)	12.4 (7.17–21.5)	11.0 (6.32–19.2)

ALL, acute lymphoid leukemia; AML, acute myeloid leukemia; BMI, body mass index; CI, confidence interval; CLL, chronic lymphocytic leukemia; CML, chronic myeloid leukemia; HL, Hodgkin’s lymphoma; IQR, interquartile range; MDS, myelodysplastic syndrome; mAb, monoclonal antibody; MM, multiple myeloma; n/a, not applicable; NHL, non-Hodgkin’s lymphoma; Ph-MPN, Philadelphia chromosome-negative myeloproliferative neoplasm. * Adjusted hazard ratios and 95% CIs were estimated with Cox proportional-hazards regression models that included age, sex, and number of comorbidities; analyses by specified comorbidity count were adjusted for age and sex. ^†^ At data cut-off, 381 of 1166 (32.7%) patients had died. ^‡^ Patients who had ever received an autologous or allogeneic stem cell transplant. ^¶^ Cancer therapy received within 30 days before COVID-19 diagnosis date. ^¶¶^ Pharmacologic therapies for COVID-19. For each therapy, the reference category included patients who did not receive the specific therapy.

**Table 3 cancers-15-01497-t003:** Baseline characteristics of, and therapy received by, patients with hematologic malignancies according to COVID-19 clinical severity.

	Patients with COVID-19 Severity Data, *N* = 1131			Odds Ratio (95% CI)	
	Total, *N* = 1131*	Mild/Moderate, *n* = 508	Severe/Critical, *n* = 623	Unadjusted	Adjusted ^†^
Age, y	*n* = 1112	*n* = 491	*n* = 621		
Median (IQR)	71 (60–79)	66 (54–76)	73 (65–82)	n/a	n/a
Age 18–49 y, *n* (%)	129 (11.6)	88 (17.9)	41 (6.6)	reference	reference
Age 50–79 y, *n* (%)	718 (64.6)	318 (64.8)	400 (64.4)	2.70 (1.81–4.02)	2.19 (1.44–3.33)
Age ≥80 y, *n* (%)	265 (23.8)	85 (17.3)	180 (29.0)	4.54 (2.89–7.14)	3.37 (2.08–5.46)
Sex, *n* (%)	*N* = 1117	*n* = 501	*n* = 616		
Female	448 (40.1)	212 (42.3)	236 (38.3)	reference	reference
Male	669 (59.9)	289 (57.7)	380 (61.7)	1.18 (0.93–1.50)	1.16 (0.90–1.50)
Comorbidity count	*N* = 1131	*n* = 508	*n* = 623		
0	305 (27.0)	181 (35.6)	124 (19.9)	reference	reference
1	469 (41.5)	199 (39.2)	270 (43.3)	1.98 (1.48–2.65)	1.53 (1.12–2.10)
≥2	357 (31.6)	128 (25.2)	229 (36.8)	2.61 (1.91–3.58)	1.62 (1.14–2.30)
Hematologic malignancy, *n* (%)	*N* = 1131	*n* = 508	*n* = 623		
Lymphoid malignancy	816 (72.1)	386 (76.0)	430 (69.0)	reference	reference
Myeloid malignancy	315 (27.9)	122 (24.0)	193 (31.0)	1.42 (1.09–1.85)	1.30 (0.98–1.72)
NHL	315 (27.9)	165 (32.5)	150 (24.1)	reference	reference
MM	258 (22.8)	130 (25.6)	128 (20.5)	1.08 (0.78–1.51)	0.97 (0.69–1.38)
CLL	169 (14.9)	55 (10.8)	114 (18.3)	2.28 (1.54–3.37)	2.02 (1.34–3.05)
HL	48 (4.2)	24 (4.7)	24 (3.9)	1.10 (0.60–2.02)	1.61 (0.82–3.16)
ALL	26 (2.3)	12 (2.4)	14 (2.2)	1.28 (0.58–2.86)	2.88 (1.22–6.82)
MDS	111 (9.8)	41 (8.1)	70 (11.2)	1.81 (1.18–2.76)	1.31 (0.82–2.11)
AML	91 (8.0)	29 (5.7)	62 (10.0)	2.35 (1.44–3.85)	3.13 (1.83–5.34)
CML	31 (2.7)	21 (4.1)	10 (1.6)	0.52 (0.24–1.15)	0.57 (0.24–1.32)
Ph-MPN	82 (7.3)	31 (6.1)	51 (8.2)	1.81 (1.10–2.98)	1.43 (0.84–2.42)
Stem cell transplantation, ^‡^ *n* (%)	*N* = 1098	*n* = 494	*n* = 604		
Autologous	100 (9.1)	65 (13.2)	35 (5.8)	0.39 (0.26–0.61)	0.56 (0.35–0.90)
Allogeneic	56 (5.1)	31 (6.3)	25 (4.1)	0.59 (0.34–1.02)	1.02 (0.56–1.84)
No	942 (85.8)	398 (80.6)	544 (90.1)	reference	reference
Cancer therapy, within 30 d, ^¶^ *n* (%)	*N* = 1131	*n* = 508	*n* = 623		
No active therapy	461 (41.8)	208 (40.9)	253 (40.6)	reference	reference
Active therapy	670 (59.2)	300 (59.1)	370 (59)	1.01 (0.80–1.29)	1.02 (0.79–1.31)
Conventional chemotherapy	255 (23)	125 (24.6)	130 (20.9)	0.86 (0.63–1.16)	1.05 (0.76–1.46)
Low-intensity chemotherapy	67 (6)	28 (5.5)	39 (6.3)	1.15 (0.68–1.92)	0.88 (0.51–1.52)
Molecular-targeted therapy	129 (11.4)	56 (11.0)	73 (11.7)	1.07 (0.72–1.59)	1.06 (0.70–1.61)
Immunotherapy, mAb only	56 (5)	24 (4.7)	32 (8.7)	1.10 (0.63–1.92)	1.21 (0.67–2.21)
Immunomodulator drugs	70 (6)	33 (6.6)	37 (5.9)	0.92 (0.56–1.53)	0.83 (0.49–1.41)
Hypomethylating agents	49 (4)	16 (3.2)	33 (5.3)	1.70 (0.91–3.17)	1.23 (0.65–2.33)
Supportive therapy	30 (2.7)	11 (2.2)	19 (3.1)	1.42 (0.66–3.05)	0.98 (0.51–1.52)
Active, not detailed	14 (1.2)	7 (1.4)	7 (1.1)	0.69 (0.21; 2.2)	0.60 (0.16–2.24)

ALL, acute lymphoid leukemia; AML, acute myeloid leukemia; BMI, body mass index; CI, confidence interval; CLL, chronic lymphocytic leukemia; CML, chronic myeloid leukemia; HL, Hodgkin’s lymphoma; IQR, interquartile range; MDS, myelodysplastic syndrome; mAb, monoclonal antibody; MM, multiple myeloma; n/a, not applicable; NHL, non-Hodgkin’s lymphoma; Ph-MPN, Philadelphia chromosome-negative myeloproliferative neoplasm. * COVID-19 clinical severity data missing for 35/1166 (3.0%) patients in the whole analysis set. Clinical severity was mild/moderate in 508/1131 (44.9%) and 623/1131 (55.1%) of patients with data available. ^†^ Adjusted odds ratios and 95% confidence intervals were estimated using multivariate logistic models that included age, sex, and comorbidity count; analyses by specified comorbidities were adjusted for age and sex. ^‡^ Patients who had ever received an autologous or allogeneic stem cell transplant. ^¶^ Cancer therapy received within 30 days before COVID-19 diagnosis date.

**Table 4 cancers-15-01497-t004:** Patients with hematologic malignancies vs. propensity-score-matched non-cancer inpatients according to time of diagnosis.

	Early Cohort			Later Cohort		
	Patients with Hematologic Malignancies *n* = 681	Non-Cancer Inpatients *n* = 5227	Odds Ratio (95% CI)	Patients with Hematologic Malignancies *n* = 215	Non-Cancer Inpatients *n* = 5312	Odds Ratio (95% CI)
Age, y				*n* = 207		
Median (IQR)	72 (62–80)	66 (53–68)	–	72 (64–80)	70 (57–81)	–
Age ≥ 65 y, *n* (%)	483 (70.9)	2829 (54.1)	2.68 (1.74–2.46)	151 (73.0)	3248 (61.1)	1.71 (1.25–2.34)
Sex, *n* (%)	*n* = 669	*n* = 5218				
Female	259 (38.7)	2213 (42.4)	reference	89 (41.4)	2272 (42.8)	reference
Male	410 (61.3)	3005 (57.6)	1.17 (0.99–1.37)	126 (58.6)	3040 (57.2)	1.06 (0.80–1.40)
Patients matched by characteristics	*n* = 669	*n* = 669		*n* = 207	*n* = 207	
Age, y						
Median (IQR)	72 (62–80)	72 (62–80)	–	72 (64–80)	72 (64–80)	–
Sex, *n* (%)						
Female	259 (38.7)	259 (38.7)	–	84 (40.6)	84 (40.6)	–
Male	410 (61.3)	410 (61.3)	–	123 (59.4)	123 (59.4)	–
Comorbidities						
Cardiac disease	148 (22.1)	420 (62.8)	0.17 (0.13- 0.21)	46 (22.2)	138 (66.7)	0.14 (0.09–0.22)
Respiratory disease	102 (15.2)	143 (21.4)	0.66 (0.50–0.88)	34 (16.4)	49 (23.7)	0.63 (0.39–1.03)
Renal disease	81 (12.1)	41 (6.1)	2.11 (1.43–3.12)	28 (13.5)	7 (3.4)	4.47 (1.91–10.5)
Diabetes	126 (18.8)	145 (21.7)	0.80 (0.62–1.05)	45 (21.7)	57 (27.5)	0.73 (0.47–1.15)
Hypertension	291 (43.5)	391 (58.4)	0.55 (0.44–0.68)	84 (40.6)	129 (62.3)	0.41 (0.28–0.61)
Pharmacologic therapies for COVID-19						
(Hydroxy)chloroquine	590 (88.2)	567 (84.8)	1.34 (0.98–1.84)	0	0	n/a
Azithromycin	318 (47.5)	343 (51.3)	0.86 (0.69–1.07)	27 (13.0)	43 (20.8)	0.57 (0.34–0.97)
Lopinavir/darunavir	383 (57.2)	424 (63.4)	0.77 (0.62–0.96)	0	0	n/a
Remdesivir	22 (3.3)	3 (0.4)	7.55 (2.25–25.3)	53 (25.6)	54 (26.1)	0.98 (0.63–1.51)
Tocilizumab	131 (19.6)	44 (6.6)	3.46 (2.41–4.96)	41 (19.8)	35 (16.9)	1.21 (0.74–2.00)
Corticosteroids	283 (42.3)	172 (25.7)	2.12 (1.68–2.67)	191 (92.3)	182 (87.9)	1.64 (0.85–3.17)
Oxygen support during COVID-19 treatment						
No or Low-flow oxygen support	511 (76.4)	575 (86.0)	reference	107 (51.7)	126 (60.9)	reference
High-flow oxygen support or mechanical ventilation	158 (23.6)	94 (14.0)	1.89 (1.43–2.51)	100 (48.3)	81 (39.1)	1.45 (0.98–2.15)
Inpatient 30-d mortality *	216/669 (32.3)	198/669 (29.6)	1.13 (0.90–1.43)	72/207 (34.8)	26/207 (12.6)	3.71 (2.25–6.13)

* Mortality proportion at 30 days. IQR, interquartile range; n/a, not applicable.

**Table 5 cancers-15-01497-t005:** Post COVID-19 condition in patients with hematologic malignancies, by patient characteristics, hematologic malignancy and COVID-19 features.

	Patients with PCC Data, *N* = 278 *			Odds Ratio (95% CI)	
	Total, *N* = 278	PCC, *n* = 76	No PCC, *n* = 202	Unadjusted	Adjusted ^†^
Age, y	*N* = 276		*n* = 200		
Median (IQR)	67 (54.5–76)	69 (54–76)	67 (54.5–75)	n/a	n/a
Age 18–49 y, *n* (%)	47 (17.0)	11 (14.5)	36 (18)	reference	reference
Age 50–79 y, *n* (%)	191 (69.2)	54 (71.1)	142 (71.0)		
Age ≥ 80 y, *n* (%)	33 (12.0)	11 (14.5)	22 (11)	1.64 (0.61–4.40)	1.24 (0.42–3.69)
Sex, *n* (%)	*N* = 271	*n* = 72	*n* = 199		
Female	149 (55.0)	42 (58.3)	107 (53.8)	reference	reference
Male	122 (45.0)	30 (41.7)	92 (46.2)	1.20 (0.70–2.08)	1.20 (0.69–2.08)
Comorbidities					
Number of 6 specified comorbidities					
0	78 (28.1)	16 (21.1)	62 (30.7)	reference	reference
≥1	200 (71.9)	60 (79.0)	140 (69.3)	1.66 (0.89–3.11)	1.57 (0.78–3.16)
Other comorbidities	144 (51.8)	44 (57.9)	100 (49.5)	1.40 (0.82–2.39)	1.36 (0.77–2.42)
Hematologic malignancy					
NHL	83 (29.9)	16 (21.1)	67 (33.2)	reference	reference
MM	66 (23.7)	21 (27.6)	45 (22.3)	1.95 (0.92–4.15)	1.86 (0.85–4.05)
CLL	29 (10.4)	10 (13.2)	19 (9.4)	2.20 (0.86–5.64)	2.10 (0.81–5.44)
HL	16 (5.8)	4 (5.3)	12 (5.9)	1.40 (0.38–4.90)	1.06 (0.25–4.41)
ALL	3 (1.1)	1 (1.3)	2 (1)	2.09 (0.18–24.5)	2.16 (0.17–27.1)
MDS	23 (8.3)	6 (7.9)	17 (8.4)	1.48 (0.50–4.35)	1.47 (0.48–4.47)
AML	28 (10.1)	8 (10.5)	20 (9.9)	1.68 (0.63–4.84)	1.58 (0.58–4.30)
CML	7 (2.5)	1 (1.3)	6 (3.0)	0.70 (0.08–6.21)	0.58 (0.06–5.31)
Ph-MPN	23 (8.3)	9 (11.8)	14 (6.9)	2.69 (0.99–7.31)	2.02 (0.68–6.02)
Stem cell transplantation ^‡^	*N* = 274	*n* = 72			
No	219 (79.9)	59 (81.9)	160 (79.2)	reference	reference
Autologous	35 (12.8)	7 (9.7)	28 (13.9)	0.68 (0.28–1.64)	0.68 (0.25–1.84)
Allogeneic	20 (7.3)	6 (8.3)	14 (6.9)	2.16 (0.43–3.17)	1.36 (0.46–4.05)
Cancer therapy, within 30 d ^§^	*N* = 276				
No active therapy	105 (38.0)	28 (36.8)	77 (38.1)	reference	reference
Active therapy	173 (62.0)	48 (63.2)	125 (61.9)	1.06 (0.61–1.82)	0.95 (0.54–1.67)
Conventional chemotherapy	62 (22.5)	17 (22.7)	45 (22.4)	1.04 (0.51–2.11)	0.99 (0.47–2.08)
Low-intensity chemotherapy	16 (5.8)	5 (6.7)	11 (5.5)	1.25 (0.40–3.92)	0.98 (0.28–3.50)
Molecular targeted therapy	34 (12.3)	11 (14.7)	23 (11.4)	1.32 (0.57–3.04)	1.03 (0.42–2.56)
Immunotherapy, mAb only	16 (5.8)	2 (2.7)	14 (7.0)	0.39 (0.08–1.84)	0.41 (0.09–1.99)
Immunomodulator drugs	19 (6.9)	4 (5.3)	15 (7.5)	0.73 (0.22–2.40)	0.63 (0.19–2.11)
Hypomethylating agents	16 (5.8)	4 (5.3)	12 (6.0)	0.92 (0.27–3.08)	0.76 (0.21–2.71)
Supportive therapy	5 (1.8)	2 (2.7)	3 (1.5)	1.83 (0.29–11.6)	1.35 (0.20–9.02)
Time period of COVID-19 diagnosis					
Early cohort (1st wave, March–June 2020)	209 (75.2)	63 (82.9)	146 (72.3)	reference	reference
Later cohort (2nd/3rd wave, July 2020–February 2021)	69 (24.8)	13 (17.1)	56 (27.7)	0.54 (0.28–1.05)	0.60 (0.30–1.20)
Care setting of COVID-19 treatment					
Outpatient	72 (25.9)	11 (14.5)	61 (30.2)	reference	reference
Inpatient	206 (74.1)	65 (85.5)	141 (69.8)	2.56 (1.26–5.18)	2.37 (1.15–4.90)
Intensive care unit	44/206 (21.4)	17/65 (26.2)	27/141 (19.1)	1.50 (0.75–2.99)	1.59 (0.75–3.31)
Pharmacologic therapies for COVID-19					
(Hydroxy)chloroquine	181 (65.1)	58 (76.3)	123 (60.9)	2.07 (1.14–3.80)	1.91 (1.03–3.53)
Azithromycin	87 (31.3)	26 (34.2)	61 (30.2)	1.20 (0.69–2.11)	0.99 (0.54–1.80)
Lopinavir/darunavir	132 (47.5)	46 (60.5)	86 (42.6)	2.07 (1.21–3.54)	2.03 (1.16–3.56)
Remdesivir	31 (11.2)	13 (17.1)	18 (8.9)	2.11 (0.98–4.55)	2.26 (1.03–4.98)
Tocilizumab	54 (19.4)	21 (27.6)	33 (16.3)	1.96 (1.05–3.66)	1.97 (1.03–3.77)
Corticosteroids	138 (49.6)	45 (59.2)	93 (46.0)	1.70 (1.00–2.90)	1.68 (0.97–2.92)
Oxygen support during COVID-19 treatment					
No	104 (37.4)	24 (31.6)	80 (39.6)	reference	reference
Low-flow oxygen support	133 (47.8)	36 (47.4)	97 (48.0)	1.24 (0.68–2.24)	1.24 (0.66–2.35)
High-flow oxygen support or mechanical ventilation	41 (14.7)	16 (21.1)	25 (12.4)	2.13 (0.98–4.64)	2.20 (0.98–4.94)
Clinical severity of COVID-19	*N* = 273	*n* = 75	*n* = 198		
Mild	69 (25.3)	14 (18.7)	55 (27.8)	reference	reference
Moderate	89 (32.6)	22 (29.3)	67 (33.8)	1.29 (0.60–2.76)	1.48 (0.66–3.37)
Severe	83 (30.4)	24 (32.0)	59 (29.8)	1.60 (0.75–3.40)	1.73 (0.77–3.90)
Critical	32 (11.7)	15 (20.0)	17 (8.6)	3.47 (1.40–8.60)	3.60 (1.35–9.62)

* PCC data were available for 278/1166 (23.8%) patients in the whole analysis set: 160 patients were excluded because time from COVID-19 diagnosis to data collection closure was <12 weeks, 324 patients were excluded because time from COVID-19 diagnosis to death was <12 weeks, and 404 patients were excluded because time from COVID-19 diagnosis to last visit was <12 weeks. PCC was present in 76/278 (27.3%) patients with data available. ^†^ Adjusted odds ratios and 95% CIs were estimated using multivariate logistic models that included age, sex, and comorbidity count. ^‡^ Patients who had ever received an autologous or allogeneic stem cell transplant. ^§^ Cancer therapy received within 30 days before COVID-19 diagnosis date. CI, confidence interval. IQR, interquartile range; n/a, not applicable; PCC, post COVID-19 condition.

## Data Availability

The datasets generated during and/or analyzed during the current study are available from the corresponding author on reasonable request.

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
