# Peer review of "COVID-19 Severity and Survival over Time in Patients with Hematologic Malignancies: A Population-Based Registry Study"

_cancers, 2023, doi:10.3390/cancers15051497_

Round 1

Reviewer 1 Report

The paper describes survival following, and severity of, COVID19, as well as post-COVID condition (4 or 6-months after), among haematological cancer patients who were not vaccinated.  Using data from a population-based register of adult patients with haematological malignancies who had positive test for COVID19 – which covered 32 centres in Madrid-, those contracting COVID in early (Feb-Jun 2020) and later waves (July 2020-Feb 2021) were compared, with follow-up to 1 March 2021; data up to Oct 2020 have been previously published.  Comparisons were also made to persons without cancer identified from another COVID register.  30- & 60-d survival were reported.

With data from the first wave having been published, there is opportunity here to examine whether there have been improvements for HM patients with COVID in later waves.  This is done to some extent but the paper could be more focused on this aspect.  For instance, it is not until section 3.5 that comparisons are highlighted (although there are some before).  The survival analysis suggests no improvement- nor are there proportionally fewer needing ICU, but there are fewer patients on the register in the later waves, and a lower proportion were admitted to hospital.  Any characteristics of patients underlying some of these changes would be interesting to assess; and could provide new insights from previous publication.  These changes should be highlighted in the summary and abstract.

In sections 3.2 and 3.3, the text is hard to follow because the denominators used in the calculations change; using the number of patients in each wave should be used in most instances.  There is no report of the percentages hospitalized in each wave in section 3.2 (from numbers in Section 3.3, in the first wave, 681 of 769 (89%) patients were hospitalized and in the later waves, 215 of 397 (54%) were); please state the number of patients seen in outpatients as well as the percentages.  Please note that the statement on lines 277-9, that the proportion of hospitalized patients [admitted to ICU] was higher in later waves than early is a function of the lower number of patients admitted to hospital- the important point is that made about the proportion of total infected patients going to ICU has not changed- which- it could be added, is despite the declines in the number of infected patients on the register, and the proportion of whom were admitted.  This point should also be modified in the discussion, lines 474-6.  Similarly in section 3.5, lines 361-6, findings are restricted to inpatients rather than the total cohort.

The numbers in the Table 1 vary depending on the variable being examined.  As Table 1 outlines the characteristics of the cohort, the authors should consider using the same patients throughout, at least restricting to those with known age and sex; exclusions for missingness could be added to Figure 1.

Introduction: Other studies suggest that mortality following COVID infection among HM patients has improved across waves – over what time period following COVID was mortality measured in these studies?

Lines 59-60 “patients with hematological patients” should be “patients with hematological malignancies”.

Summary and abstract refer to different numbers of comorbidities (3+ in former referring to mortality, 1+ in latter for COVID severity).  Table 1 describes frequencies with 0, 1 and 2+; table 3 (survival) on 0,1,2 and 3+.  What is the rationale for the different groupings?

Abstract, penultimate sentence: among prognostic factors for COVID-19 condition, hospitalisation is listed.  Is this not expected (& not prognostic)?

Line 196 six specified comorbidities- are these the same conditions listed on lines 193-194?

“Comorbidity count”- number of comorbidities better?

Line 236 It is stated that thirty-one hospitals are included, when earlier the number of centres is 32?

Table 1 is quite long- it would be sufficient to report age as median and IQRs, categories are not necessary.  Under Cancer therapy, give the numbers receiving chemotherapy, with specific types listed below, to be consistent with the text.

Table 3- please state in the title whether the HRs refer to 60-d or 30-d survival.  There is no reference group listed for the COVID therapies in this table or in tables 4 and 5.

Author Response

The paper describes survival following, and severity of, COVID19, as well as post-COVID condition (4 or 6-months after), among haematological cancer patients who were not vaccinated.  Using data from a population-based register of adult patients with haematological malignancies who had positive test for COVID19 – which covered 32 centres in Madrid-, those contracting COVID in early (Feb-Jun 2020) and later waves (July 2020-Feb 2021) were compared, with follow-up to 1 March 2021; data up to Oct 2020 have been previously published.  Comparisons were also made to persons without cancer identified from another COVID register.  30- & 60-d survival were reported.

With data from the first wave having been published, there is opportunity here to examine whether there have been improvements for HM patients with COVID in later waves.  This is done to some extent but the paper could be more focused on this aspect.  For instance, it is not until section 3.5 that comparisons are highlighted (although there are some before).  

   Response to Reviewer 1, comment 1:

We are grateful to the reviewer for the thorough reading of the manuscript and for the helpful comments and suggestions. We agree that the comparison between periods should be prioritized.

- The comparison between early and later cohorts is now included after the flow-diagram, section 3.3.

The survival analysis suggests no improvement- nor are there proportionally fewer needing ICU, but there are fewer patients on the register in the later waves, and a lower proportion were admitted to hospital.  Any characteristics of patients underlying some of these changes would be interesting to assess; and could provide new insights from previous publication.  These changes should be highlighted in the summary and abstract.

   Response to Reviewer 1, comment 2:

Patient’s characteristics in both periods are described and compared in detail in Supplementary Table 1. In the first wave, patients were older and had more comorbidities; in the second wave, more patients had received a SC transplantation and the proportion of hospitalized patients was lower. Among inpatients, the proportion of critical clinical status, high-flow oxygen support or mechanical ventilation, and intensive care admission was higher in the second/third wave than in the first wave.

- We have re-drafted, shortened (word count=200) and focused the abstract on the comparison between early sand later cohorts, particularly in hospitalized patients:

- Abstract: Mortality rates for COVID-19 have declined over time in the general population, but data in patients with hematologic malignancies are contradictory. We identified independent prognostic factors for COVID-19 severity and survival in unvaccinated patients with hematologic malignancies, compared mortality rates over time and versus non-cancer inpatients, and investigated post COVID-19 condition. Data were analyzed from 1166 consecutive, eligible patients with hematologic malignancies and COVID-19 prior to vaccination roll-out from the population-based HEMATO-MADRID registry, Spain, stratified into early (February-June, 2020; n=769 (66%)) and later (July 2020-February 2021; n=397 (34%)) cohorts. Propensity-score matched non-cancer inpatients were identified from the SEMI-COVID registry. Patients in the later cohort were more likely to receive outpatient care (45.8% vs 11.4%, OR 6.55, 95% CI: 4.87-8.82). The proportion of hospitalized patients admitted to the ICU was higher in the later cohort (103/215, 47.9%) compared with the early cohort (170/681, 25.0%, OR 2.77; 2.01-3.82). The reduced 30-day mortality between early and later cohorts of non-cancer inpatients (29.6% vs 12.6%, OR 0.34 [0.22-0.53]) was not paralleled in inpatients with hematologic malignancies (32.3% vs 34.8%, odds ratio 1.12 [0.81-1.5]). Among evaluable patients, 27.3% had post COVID-19 condition. These findings will help inform evidence-based preventive and therapeutic strategies for patients with hematologic malignancies.

In sections 3.2 and 3.3, the text is hard to follow because the denominators used in the calculations change; using the number of patients in each wave should be used in most instances.  There is no report of the percentages hospitalized in each wave in section 3.2 (from numbers in Section 3.3, in the first wave, 681 of 769 (89%) patients were hospitalized and in the later waves, 215 of 397 (54%) were); please state the number of patients seen in outpatients as well as the percentages.  

Response to Reviewer 1, comment 3:

Agree. We now stated in 3.2 the number of patients seen in outpatients as well as the percentages.

- Patients in the later cohort were more likely to receive outpatient care (88/769, 11.4% vs 182/397, 45.8%, early vs later cohort, OR 6.55, 95% CI: 4.87-8.82).

Please note that the statement on lines 277-9, that the proportion of hospitalized patients [admitted to ICU] was higher in later waves than early is a function of the lower number of patients admitted to hospital- the important point is that made about the proportion of total infected patients going to ICU has not changed- which- it could be added, is despite the declines in the number of infected patients on the register, and the proportion of whom were admitted.  This point should also be modified in the discussion, lines 474-6.  Similarly in section 3.5, lines 361-6, findings are restricted to inpatients rather than the total cohort.

  Response to Reviewer 1, comment 4:

Agree. We improved clarity in the sentences of the results section (previously, 277-9) and in the corresponding discussion section (previously, 474-6).
(previously, 358-360): analysis in the total population; (previously, 361-366): analysis in inpatients)

- Results (previously, 277-79): The overall proportion of patients admitted to an ICU was similar between cohorts (22.1% vs 25.9%, OR 1.25; 95% CI: 0.94-1.66). The proportion of hospitalized patients admitted to the ICU was higher in the later cohort (103/215, 47.9%) compared with the early cohort (170/681, 25.0%, OR 2.77; 95% CI: 2.01-3.82).  

- Discussion: In our study, we found some differences in patient characteristics between the later and early cohorts that may help explain the lack of survival difference between waves. Although the proportion of ICU admissions was similar in both total cohorts, among hospitalized HM patients, the proportions of patients with critical COVID-19 and of ICU admissions was higher in the later cohort.

The numbers in the Table 1 vary depending on the variable being examined.  As Table 1 outlines the characteristics of the cohort, the authors should consider using the same patients throughout, at least restricting to those with known age and sex; exclusions for missingness could be added to Figure 1. 

      Response to Reviewer 1, comment 5:

We understand the Reviewer’s comment. We considered that there was not a good solution to get the same denominator for all variables in Table 1. Even taking the denominator of those without age and sex data missing, the other variables would have different denominators.
Of note, the variable with maximum missing data has n=39/1166, 3.3%. We consider the likelihood of introducing a bias with such a small proportion of missing data is limited.

Introduction: Other studies suggest that mortality following COVID infection among HM patients has improved across waves – over what time period following COVID was mortality measured in these studies?

      Response to Reviewer 1, comment 6:

This is an important comment. All three studies that analyzed trends in mortality in HM patients compared the first and the second/third waves of the pandemic. But median follow-up time after COVID-19 infection is specified in none of them. From the Figure representing survival curves, we infer that median follow-up time in both cohorts is much shorter in other studies than in ours (At data cut-off, median follow-up was 40 days, IQR, 16-99).

- We now included a sentence in the discussion section regarding follow-up time after COVID infection on line 473: A shorter follow-up time in other studies could also contribute to a lower mortality.

Summary and abstract refer to different numbers of comorbidities (3+ in former referring to mortality, 1+ in latter for COVID severity).  Table 1 describes frequencies with 0, 1 and 2+; table 3 (survival) on 0,1,2 and 3+.  What is the rationale for the different groupings?

      Response to Reviewer 1, comment 7:

As correctly pointed, the total number of comorbidities associated with severity and mortality was different (1+ for severity, 3+ for mortality). The number of comorbidities in Tables 3 is extended to 3 to show the minimum number of comorbidities associated with mortality. Table 1 was descriptive, we considered that 0, 1, 2 comorbidities was informative enough. Table 2 showed associations, we considered that as 1 comorbidity present was already associated, grouping 2+ did not imply a loss of information.

Abstract, penultimate sentence: among prognostic factors for COVID-19 condition, hospitalisation is listed.  Is this not expected (& not prognostic)?

     Response to Reviewer 1, comment 8:

Hospitalization is used here as a proxy of severity. We considered worthwhile underlying that severity/hospitalization increased the likelihood of post COVID-19 condition although this outcome is also reported in not hospitalized patients.

Line 196 six specified comorbidities- are these the same conditions listed on lines 193-194?

 A comorbidity count was determined based on presence of the six comorbidities prespecified above.

“Comorbidity count”- number of comorbidities better?

    Response to Reviewer 1, comment 9:

Agree. We improved the wording for disambiguation purpose.

- In Line 195: A comorbidity count was determined based on presence of the six comorbidities prespecified above.

Thank you for the suggestion. We used ‘comorbidity count’ for the number of comorbidities among the number of 6 prespecified comorbidities (cardiac disease, pulmonary disease not including lung cancer, renal disease, diabetes, hypertension, and body mass index [BMI] ≥35). In line with the literature and for limiting the length of the labels in the tables, we propose to keep ‘comorbidity count’ when applicable.

- Following Reviewer 3 comment, we dropped detailed information by comorbidity in the tables.

Line 236 It is stated that thirty-one hospitals are included, when earlier the number of centres is 32?

   Response to Reviewer 1, comment 10:
32 centers corresponded to the number of affiliated centers. 31 is the number of centers who reported.

- For disambiguation purpose, and following Reviewer 1 suggestion, we changed 32 to 31 in line 161.

Table 1 is quite long- it would be sufficient to report age as median and IQRs, categories are not necessary.  Under Cancer therapy, give the numbers receiving chemotherapy, with specific types listed below, to be consistent with the text.

  Response to Reviewer 1, comment 11:
Agree.

- Age categories were dropped from Table 1;
- Under Cancer therapy in Table 1, the total numbers receiving chemotherapy were added in the category ‘Active therapy’ to be consistent with the text.

Table 3- please state in the title whether the HRs refer to 60-d or 30-d survival.  There is no reference group listed for the COVID therapies in this table or in tables 4 and 5.

Response to Reviewer 1, comment 12:
In Table 3, Hazard ratios (HR) do not refer specifically to a point in time but to the total follow-up time.

- In tables 3, we included the following sentence in the footnote:
¶¶Pharmacologic therapies for COVID-19. For each therapy, the reference included patients who did not receive the specific therapy.

Reviewer 2 Report

Martinez-Lopez et al., evaluated the morbidity and mortality in patients with hematologic malignancies and were infected with COVID-19 in the article, "COVID-19 severity and survival over time in patients with hematologic malignancies: a population-based registry study". The authors should be commended for the effort and they mentioned it is one of the largest cohort studies in this context. The article is very well written. In general, I have a few comments- many studies have shown similar results till date and this study confirms the previously establish findings. Also, it is not a surprise to see the findings given the cytopenias the patients with hematologic malignancies experience be it from the underlying malignancy or due to chemotherapy administered to treat the malignancy. In addition, many patients with hematologic malignancies also receive B-cell immunosuppressive therapy, which may impair immune system. Wondering if the authors made any attempt on evaluating any of these factors contributing to worse morbidity and mortality for COVID19 in patients with hematologic malignancies. 

Author Response

Martinez-Lopez et al., evaluated the morbidity and mortality in patients with hematologic malignancies and were infected with COVID-19 in the article, "COVID-19 severity and survival over time in patients with hematologic malignancies: a population-based registry study". The authors should be commended for the effort and they mentioned it is one of the largest cohort studies in this context. The article is very well written. In general, I have a few comments- many studies have shown similar results till date and this study confirms the previously establish findings. Also, it is not a surprise to see the findings given the cytopenias the patients with hematologic malignancies experience be it from the underlying malignancy or due to chemotherapy administered to treat the malignancy. In addition, many patients with hematologic malignancies also receive B-cell immunosuppressive therapy, which may impair immune system. Wondering if the authors made any attempt on evaluating any of these factors contributing to worse morbidity and mortality for COVID19 in patients with hematologic malignancies. 

Response to Reviewer 2, comment 1:

We thank the reviewer for his comments and agree with the high relevance of exploring the role of cytopenia and B-cell immunosuppression in morbidity and mortality. Unfortunately, the registry data collection was not granular enough on laboratory data to address this important questions.

Reviewer 3 Report

Cancers-2101980

Title: COVID-19 severity and survival over time in patients with hematologic malignancies: a population-based registry study

The Authors presented an extensive analysis of the populationbased registry identified independent prognostic factors for COVID19 severity and survival in 1166 unvaccinated patients with hematologic malignancies, compared mortality rates over time and versus noncancer inpatients, and investigated post COVID19 condition.

The Authors referred to a 2020 meta-analysis on a comparable topic (Vijenthira A, et al. Blood. 2020). However, one of the main questions raised in the paper was whether increased clinical experience on the course and treatment of COVID19 has changed the outcomes of patients with haematological malignancies.

The great advantage of the work is its prospective and multicenter character. The manuscript is carefully written, referring to the latest and high-quality literature. Nevertheless, the Reviewer would like to add some comments.

1. Abstract: „Prognostic factors analysis was performed with Cox proportionalhazards and logistic regression models.” - information on the method of statistical analysis in the "Abstract" section is redundant.

2. Due to the large amount of data in Tables 1, 2, 3 and 5, they become illegible. I suggest:

- limit age subgroups to 3: 18-49y, 50-79y and ≥80y;

- remove individual Comorbidities and keep only data with number of Comorbidities (0, 1, ≥2).

3. The publication lacks detailed data on oncological treatment. The reader is interested in what "low-intensity chemotherapy", "hypomethylating agents", "supportive therapy" mean. Among how many patients what therapies were combined. Wouldn't a more readable information be the number of NEUTs in the morphology?

4. The manuscript lacks detailed data on COVID-19 therapy, and this seems to be the key information - what form of therapy (with exact dosage and number of days of treatment) turned out to be the most useful for patients.

Author Response

The Authors presented an extensive analysis of the population‑based registry identified independent prognostic factors for COVID‑19 severity and survival in 1166 unvaccinated patients with hematologic malignancies, compared mortality rates over time and versus non‑cancer inpatients, and investigated post COVID‑19 condition.

The Authors referred to a 2020 meta-analysis on a comparable topic (Vijenthira A, et al. Blood. 2020). However, one of the main questions raised in the paper was whether increased clinical experience on the course and treatment of COVID‑19 has changed the outcomes of patients with haematological malignancies.

The great advantage of the work is its prospective and multicenter character. The manuscript is carefully written, referring to the latest and high-quality literature. Nevertheless, the Reviewer would like to add some comments.

  1. Abstract: „Prognostic factors analysis was performed with Cox proportional‑hazards and logistic regression models.” - information on the method of statistical analysis in the "Abstract" section is redundant.

Response to Reviewer 3, comment 1:
Agree. Text on methods of statistical analysis dropped from the abstract text.

  1. Due to the large amount of data in Tables 1, 2, 3 and 5, they become illegible. I suggest:

- limit age subgroups to 3: 18-49y, 50-79y and ≥80y;

- remove individual Comorbidities and keep only data with number of Comorbidities (0, 1, ≥2).

Response to Reviewer 3, comment 2:
Agree. Age groups and comorbidities were reduced in the tables.

  1. The publication lacks detailed data on oncological treatment. The reader is interested in what "low-intensity chemotherapy", "hypomethylating agents", "supportive therapy" mean. Among how many patients what therapies were combined. Wouldn't a more readable information be the number of NEUTs in the morphology?

Response to Reviewer 3, comment 3:

Agree. We now provide in the text further examples of therapies included in the oncological treatments groups. We do not have information on combined therapies. We are unsure about the meaning of the last sentence.

- ‘Active’ antineoplastic treatment was defined as patients having received anticancer therapy within 30 days prior to COVID-19 diagnosis; therapies were classified as conventional chemotherapy (Includes intensive and standard dosing), low-intensity chemotherapy (includes Single-agent hydroxyurea, chlorambucil, or cyclophosphamide), hypomethylating agents (Includes azacitidine and decitabine), monoclonal antibodies (Includes anti-CD20 and anti-CD38), immunomodulatory drugs (Includes lenalidomide, pomalidomide, and thalidomide), molecular-targeted therapies (Includes tyrosine kinase inhibitors, Bruton's tyrosine kinase inhibitors, Aurora kinase inhibitors, PI3K inhibitors, proteasome inhibitors, and histone deacetylase inhibitors), or supportive care (Includes transfusion and hematopoietic growth factor support).

  1. The manuscript lacks detailed data on COVID-19 therapy, and this seems to be the key information - what form of therapy (with exact dosage and number of days of treatment) turned out to be the most useful for patients.

Response to Reviewer 3, comment 4:

We thank the reviewer for his comments and agree on the need to assess detailed information on the therapies received by our patients. Unfortunately the registry data collection does not allow to go further and analyze therapies dosage/days.

Round 2

Reviewer 1 Report

Thank you for addressing my comments.

There is one remaining comment that requires attention.  In the abstract, it is stated that among hospitalised patients, a higher proportion were admitted to ICU in the later than earlier cohort- this must be tempered with the fact that a lower proportion of patients were hospitalised in the later waves (215 of 397 (54%) in the later compared to 681 of 769 (89%) in the earlier).   Similar text should also be added to the results and the discussion.

Author Response

Point-by-point response:

Reviewer's comment: 

There is one remaining comment that requires attention.  In the abstract, it
is stated that among hospitalised patients, a higher proportion were admitted
to ICU in the later than earlier cohort- this must be tempered with the fact
that a lower proportion of patients were hospitalised in the later waves (215
of 397 (54%) in the later compared to 681 of 769 (89%) in the earlier).   
Similar text should also be added to the results and the discussion.

Modification in the text following Reviewer’s suggestion:

Abstract

Patients in the later cohort were more likely to receive outpatient care (45.8% vs 11.4%, OR 6.55, 95% CI: 4.87-8.82).
A lower proportion of patients were hospitalized in the later waves (54.2%) compared to the earlier (88.6%), OR 0.15, 95%CI 0.11-0.20. The proportion of hospitalized patients admitted to the ICU was higher in the later cohort (103/215, 47.9%) compared with the early cohort (170/681, 25.0%, 2.77; 2.01-3.82).

Results: 3.2. COVID-19 diagnosis and treatment

Patients in the later cohort were more likely to receive outpatient care (45.8% vs 11.4%, OR 6.55, 95% CI: 4.87-8.82).

… A lower proportion of patients were hospitalized in the later waves (54.2%) compared to the earlier (88.6%), OR 0.15, 95%CI 0.11-0.20. The overall proportion of patients admitted to an ICU was similar between cohorts (22.1% vs 25.9%, OR 1.25; 95% CI: 0.94-1.66). The proportion of hospitalized patients admitted to the ICU was higher in the later cohort (103/215, 47.9%) compared with the early cohort (170/681, 25.0%, OR 2.77; 95% CI: 2.01-3.82).

Reviewer 3 Report

I regret that the Authors have not included information on the treatment of COVID-19, but I understand that this knowledge is not available. I have no further comments on the article.

Author Response

We thank the reviewer for his understanding of the limited detailed data availability on Covid-19 therapies.